Internet-delivered therapist-guided physical activity for mild to moderate depression: a randomized controlled trial

Ström Morgan 1
Uckelstam Carl-Johan 1
Andersson Gerhard 2 3
Hassmén Peter 1
Umefjord Göran 4
Carlbring Per 5 per@carlbring.se
1 Department of Psychology, Umeå University , Sweden
2 Department of Clinical Neuroscience, Karolinska Institutet , Stockholm , Sweden
3 Department of Behavioural Sciences and Learning, Psychology, Swedish Institute for Disability Research, Linköping University , Sweden
4 Department of Public Health and Clinical Medicine, Umeå University , Sweden
5 Department of Psychology, Stockholm University , Sweden
Black Kevin
Electronic publication date: 2013 Oct 3
Publication date: 2013
Volume: 1
Electronic Location ID: e178
Received 2013 Jul 19; Accepted 2013 Sep 16
Copyright: © 2013 Ström et al.
Copyright year: 2013
Copyright holder: Ström et al.
License: This is an open access article distributed under the terms of the Creative Commons Attribution License, which permits unrestricted use, distribution, and reproduction in any medium, provided the original author and source are credited.
License URL: https://creativecommons.org/licenses/by/3.0/

Keywords: Physical activity, Depression, Internet, Psychotherapy, Treatment, RCT

Funding: Swedish Council for Working Life and Social Research Swedish Research Council This study was made possible by a generous grant from the Swedish Council for Working Life and Social Research, Swedish Research Council, and a professor’s contract awarded to Gerhard Andersson. The funders had no role in study design, data collection and analysis, decision to publish, or preparation of the manuscript.

==============================
Objective. The main hypothesis, and the objective of the study, was to test if the participants allocated to the treatment group would show a larger reduction in depressive symptoms than those in the control group.

Methods. This study was a randomized nine week trial of an Internet-administered treatment based on guided physical exercise for Major Depressive Disorder (MDD). A total of 48 participants with mild to moderate depression, diagnosed using the Structured Clinical Interview for DSM-IV Axis I Disorders, were randomized either to a treatment intervention or to a waiting-list control group. The main outcome measure for depression was the Beck Depression Inventory-II (BDI-II), and physical activity level was measured using the International Physical Activity Questionnaire (IPAQ). The treatment program consisted of nine text modules, and included therapist guidance on a weekly basis.

Results. The results showed significant reductions of depressive symptoms in the treatment group compared to the control group, with a moderate between-group effect size (Cohen’s d = 0.67; 95% confidence interval: 0.09–1.25). No difference was found between the groups with regards to increase of physical activity level. For the treatment group, the reduction in depressive symptoms persisted at six months follow-up.

Conclusions. Physical activity as a treatment for depression can be delivered in the form of guided Internet-based self-help.

Trial Registration. The trial was registered at ClinicalTrials.gov (NCT01573130).

Introduction

The World Health Organization (World Health Organization, 2001) predicts that depression will be one of the three most burdensome diseases in the world in 2020. With the advancement of communication technologies, new ways of providing and delivering psychological treatments have emerged (Carlbring & Andersson, 2006). The Internet has made it possible to reach people over great distances and provide psychological interventions to a vast number of patients at a low cost due to shorter treatment time per person (Andersson, 2009). Internet-delivered treatments also have the opportunity to increase accessibility for patients in remote geographical locations and to make support available for people who would not otherwise seek care (Newman et al., 2011). Furthermore, Internet-delivered treatments have the possibility of giving patients quick feedback and presentation of material in a step-by-step basis (Titov, 2011).

Currently, several studies have investigated the effects of Internet-delivered treatments for depression (Johansson & Andersson, 2012). A large part of these studies have been based on cognitive behavior therapy (CBT) as the main theoretical framework, but there are exceptions (Johansson et al., 2012). Andersson & Cuijpers (2009) did a meta-analysis and found a significant difference between supported (d = 0.61) and unsupported (d = 0.25) depression treatments. In a more recent meta-analysis, a similar result was found by Richards & Richardson (2012). In addition, Johansson & Andersson (2012) found a strong and significant association between support and effect size with a Spearman correlation of ρ = 0.64, indicating that more support yields larger effects.

Since the beginning of the 20th century, a large amount of research has been conducted concerning the effects of physical activity on clinical depression. Several studies have found treatment effects ranging in size from moderate to large (Barbour, Edenfield & Blumenthal, 2007; Silveira et al., 2013). However, there is still no consensus about the mechanisms of change that mediate reductions in depressive symptoms following psychotherapy (Lundh, 2009), and there is also limited knowledge regarding mediators of change following physical activity for depression, apart from the physiological effects of increased activity.

A meta-analysis by Davies et al. (2012) investigated the effects of Internet-delivered interventions to increase physical activity levels. The result showed generally small but statistically significant increases in physical activity levels in the 34 studies reviewed. To our knowledge, there are few studies, if any, on guided Internet-delivered physical activity for major depression.

The purpose of the present study was to evaluate a treatment for major depression based on physical exercise administered via the Internet. The treatment program was intended to decrease depressive symptoms and to motivate participants to increase their level of physical activity. The treatment group was compared to a waiting-list control group.

The main hypothesis, and the objective of the study, was to test if the participants allocated to the treatment group would show a larger reduction in depressive symptoms than those in the control group, as measured by the Beck Depression Inventory: Second Version (BDI-II; Beck et al., 1988) and the Montgomery-Åsberg Depression Rating Scale: Short Version (MADRS-S; Svanborg & Åsberg, 2001). Since depression also has impact on other aspects of a person’s life we wanted to explore this further (Kennedy, Eisfeld & Cooke, 2001). Hence, it was hypothesized that participants in the treatment group would show larger reductions of anxiety symptoms as measured with the Beck Anxiety Inventory (BAI; Beck et al., 1988), and greater increases in levels of physical activity as measured with the International Physical Activity Questionnaire (IPAQ; Craig et al., 2003) compared to participants in the control group. Moreover, the authors expected that participants in the treatment group would show a larger increase in quality of life compared to the control, measured with the Quality of Life Inventory (QOLI; Frisch et al., 1992).

Method

Information about the study was advertised in a major Swedish newspaper, online with Google AdWords, and at an online service site containing information about ongoing research projects on Internet therapy. The participants were recruited between January and February 2012. The basic inclusion criteria in the study were mild to moderate major depression diagnosis and a sedentary lifestyle. Exclusion was based on the following criteria: subclinical depressive symptoms, severe depressive symptoms, dysthymia as a primary diagnosis, elevated suicide risk, high levels of physical activity prior to treatment, recent changes in medication and/or somatic illness making physical exercise inappropriate.

The eligibility screening process consisted of self-report questionnaires regarding depressive symptoms, anxiety, and level of physical activity, and a clinical interview via telephone to investigate the primary diagnoses of the participants. The screening questionnaires used were the MADRS-S, BAI, and IPAQ. The telephone interview was based on the Structured Clinical Interview for DSM-IV: Clinical version (SCID-I-CV; First et al., 1997). A full description of participant recruitment is included in Fig. 1, and a demographic description of the participants at pre-treatment is presented in Table 1.

Figure 1 Flowchart of study participants.

Table 1 Demographic description of the participants at pre-treatment.

	Treatment (n = 24)	Control (n = 24)	Total (n = 48)	
Sex	
Female	20	(83.3%)	20	(83.3%)	40	(83.3%)	
Male	4	(16.7%)	4	(16.7%)	8	(16.7%)	
Age	
Mean (SD)	48.8	(12.7)	49.6	(8.7)	49.2	(10.7)	
Min–Max	24–67		35–65		24–67		
Marital status	
Married/Living together	13	(54.2%)	9	(37.5%)	22	(45.8%)	
Living apart	2	(8.3%)	2	(8.3%)	4	(8.3%)	
Single	8	(33.3%)	13	(54.2%)	21	(43.8%)	
Other	1	(4.2%)	0		1	(2.1%)	
Highest educational level	
Compulsory school	1	(4.2%)	0		1	(2.1%)	
Secondary school	3	(12.5%)	3	(12.5%)	6	(12.5%)	
Vocational school	0		3	(12.5%)	3	(6.3%)	
College/university (on-going)	2	(8.3%)	2	(8.3%)	4	(8.3%)	
College/university (compl.)	18	(75%)	16	(66.7%)	34	(70.8%)	
Medication	
None	10	(41.7%)	14	(58.3%)	24	(50.0%)	
Earlier	11	(45.8%)	6	(25.0%)	17	(35.4%)	
Present	3	(12.5%)	4	(16.7%)	7	(14.6%)	
Psychotherapy	
None	9	(37.5%)	9	(37.5%)	18	(37.5%)	
Earlier	15	(62.5%)	15	(62.5%)	30	(62.5%)	
Present	0		0		0		

Table 2 Overview of content and home assignments of the intervention.

Module	Content	Home assignments	
1.	Brief introduction to depression; signs and symptoms. Explanation of depression using the spiral modela. Overview of different types of PAb, and how PA can be helpful to treat depression. Presentation of treatment structure and how to use the EWBSc. Presentation of how to set up and use the pedometer.	1. Participants are asked to give a brief narrative of their depressive symptoms and prior experience of PA.
2. What are the participants’ thoughts about their chance of increasing PA level?
3. Which are the participants’ main barriers for PA?
4. The participants are asked to use the pedometer in three walks during the coming week and to register the total number of steps taken.	
2.	Introduction on how a sedentary lifestyle influences overall health. Description of how PA affects the human body in physical, mental, and neurological ways, and level of PA needed to acquire positive health effects. Presentation of stages of changed. Most common barriers for PA and how to overcome them. Examination of pros and cons of increasing PA or maintaining the status quo, using a motivational balance exercise.	1. What is the participant’s view of the treatment program so far?
2. Which stage of change do the participants find themselves in right now?
3. What do the participants believe is their main obstacle for PA?
4. Motivational balance exercise.
5. Participants are encouraged to keep taking three walks the coming week and if they want, to increase the length of the walks.	
3.	Introduction to goal setting using SMART goal setting principlese. How to work with activity scheduling to incorporate regular PA into everyday life. Examples of different forms of PA. Important aspects of change management and how to increase self-efficacy for PA.	1. SMART goal-setting for the next week.
2. Making a schedule of PA activities to meet the goals for next week.
3. Patients are encouraged to register all PA conducted during the week (using the pedometer).	
4.	Introduction on how to follow up and review the goal and schedule from last week. Possible links between PA and mood.	1. From this week on, SMART goal-setting, activity scheduling and registration of PA for the coming week are incorporated as a weekly routine.
2. Participants are asked to review the goal and the schedule from the past week.
3. Which were the biggest obstacles for PA this week?
4. How did the participants deal with them?	
5.	Introduction to handling setbacks and relapses during behaviour change. Presentation of the most common thinking errors when afflicted by setbacks and how to deal with them. How to reward progress in PA and to facilitate long lasting behaviour change.	1. Do the participants recognize any of the common thinking errors when afflicted by setbacks?
2. What progress have the participants experienced so far?
3. How can these accomplishments be rewarded?	
6.	How to get sufficient rest and recovery after PA. General nutrition advice before and after PA.	1. What do the participants find particularly important as a “take home message” regarding rest and nutrition?	
7.	Participants are introduced to aspects of acceptance and commitment theoryf, and are initiated to think about how PA can be part of heightened quality of life. Subjects learns about living in accordance with what they value in terms of health and PA, and are introduced to the concept of having a permissive attitude towards all experiences when moving in their valued direction, even the difficult ones.	1. Valued direction exercise focusing on health and PA.	
8.	Mindfulness walking and how to incorporate acceptance in the struggle to increase and maintain PA	1. Participants are encouraged to do a mindfulness walking exercise. What were their experiences?	
9.	How to maintain PA after the end of the treatment program. Summary of the previous modules.	1. Participants are encouraged to answer the post-treatment questionnaires administered over the Internet.	
Notes.

a Haase and coworkers (2010).

b Physical activity.

c Encrypted web-based system.

d Prochaska & DiClemente (1983).

e Hassmén & Hassmén (2005).

f Hayes et al. (2006).

Table 3 Results at pre- and post-treatment for measures of depression, anxiety, physical activity, and quality of life.

In addition, the 6-month follow-up for the treatment group is reported.

Time	Treatment (n = 24)	Control (n = 24)	ANOVA	Between-groups effect size	Within-groups effect size	
	M	SD	M	SD	F	Cohen’s d	Cohen’s d	
Montgomery-Åsberg Depression Rating Scale: Self-Rated Version (MADRS-S)	
Pre	23.54	(4.39)	23.92	(3.87)	G: 3.02	0.62	Tx: 1.30	
Post	15.71	(7.54)	20.38	(7.87)	T: 29.82***		C: 0.58	
6-mo	14.46	(7.63)	N/a	N/a	I: 4.25*			
Beck Depression Inventory: Second Version (BDI-II)	
Pre	26.92	(9.30)	28.25	(7.08)	G: 2.52	0.67	Tx: 0.89	
Post	17.88	(11.30)	24.04	(6.86)	T: 48.77***		C: 0.62	
6-mo	15.63	(11.44)	N/a	N/a	I: 6.49*			
Beck Anxiety Inventory (BAI)	
Pre	15.50	(7.96)	15.71	(6.53)	G: 0.08	0.14	Tx: 0.37	
Post	12.92	(6.36)	13.71	(5.27)	T: 9.29**		C: 0.34	
6-mo	10.71	(6.41)	N/a	N/a	I: 0.15			
International Physical Activity Questionnaire (IPAQ)	
Pre	778	(695)	953a	(670)a	G: 0.00	0.20	Tx: 0.66	
Post	1331	(990)	1143	(918)a	T: 6.41*		C: 0.24	
6-mo	1282	(1255)	N/a	N/a	I: 1.47			
Quality of Life Inventory (QOLI)	
Pre	−0.50	(1.72)	−0.25	(1.54)	G: 0.12	0.04	Tx: 0.36	
Post	0.16	(1.99)	0.23	(1.47)	T: 11.49**		C: 0.33	
6-mo	0.33	(2.00)	N/a	N/a	I: 0.28			
Notes.

G Group effect

T Time effect

I Interaction effect

Tx Treatment group

C Control group

N/a not available

*** p < .001.

** p < .01.

* p < .05.

a n = 22 due to incomprehensible data.

The study included 48 participants meeting the criteria for major depression. The participants were randomly allocated to the two groups, treatment or waiting list control, by a person independent of the research group using a true random number service on the Internet (www.random.org). Participants’ levels of depression as well as anxiety, physical activity, and quality of life were measured at pre- and post-treatment, as well as at a 6-months follow-up. The study was conducted between February and April 2012, and the follow-up measurements were collected in October 2012.

The main outcome measure for the assessments of depression and depressive symptoms was the BDI-II with reported reliability estimates (coefficient α) for Swedish samples between α = .88 and α = .92 (Carlbring et al., 2007). As a secondary measure of depression, we used the MADRS-S, which has a reported reliability estimate of α = .84 (Fantino & Moore, 2009). The psychometric properties of the Internet versions of these instruments have proved to be equivalent to the paper-and-pencil versions (Holländare, Andersson & Engström, 2010). For evaluating changes in anxiety, physical activity, and quality of life, the BAI, IPAQ, and QOLI were administered; all of which showed satisfactory psychometric properties (Lindner et al., 2013; Ekelund et al., 2006).

Intervention

The treatment used in this study was a guided self-help program administered through an Internet-based system. The program consisted of nine text modules developed by the authors, consisting of 72 pages in total. Participants were given one text module every Monday over nine weeks. At the end of each week, the therapists gave written feedback on home assignments included in the modules. Message exchange and the delivery of the treatment modules used were transmitted via an encrypted web-based system. Some material, such as the pedometer given to participants in the treatment condition, and the written form of consent, were sent using the postal service. The purpose of the pedometer was for the participants to monitor their own physical activity in the form of walking, which was part of the treatment program. The pedometers were used for motivational purposes alone and since participants had different ways of using them no data was collected in the study.

The modules consisted of self-help texts about how to become more physically active (see Table 2). Each module ended with 3–5 essay questions where the participant was asked to report on the progress and the weekly planning of the exercise. The core principles of the program were inspired by Haase and coworkers (2010) and were intended to: (1) maximize the likelihood for participants to increase and maintain physical activity; (2) maximize the likelihood of participants remaining engaged in the program; (3) focus on the participants’ preferences and needs, taking particular notice of the challenges faced by people with major depression; (4) promote physical activity in a broad sense in accordance with the WHO’s guidelines for physical activity (Mendes, Sousa & Barata, 2011), including all types of activity in everyday life; (5) increase self-efficacy for physical activity; and (6) help participants to master challenges faced with when trying to get more active. The study was examined and approved by the regional ethical committee at Umeå, Sweden (2011-145-31 Ö), and is registered at ClinicalTrials.gov: NCT01573130.

Statistical Analysis

The data were analyzed using the statistical programming language R, version 3.0.0. Reproducible code can be found in the supplemental file (Progredi.R). Significance testing of group differences regarding demographic data in Table 1 and pre-treatment measurements was conducted using Welch’s two sample t-test for continuous data and chi-2-tests for nominal data.

Differences between the groups in pre- and post-measurements were analyzed using a two-way mixed analysis of variance model (ANOVA), with treatment condition and time used as independent variables. Effect sizes were calculated using Cohen’s d.

Data were analyzed using an intention-to-treat (ITT) approach. The response rate for all outcome measures was 100% (48 of 48) for pre- and post-treatment measurements, and 87.5% (21 of 24) for the 6-month follow-up. Due to the small number of missing data, we did not impute the missing data at 6-month follow-up, but used baseline carried forward as an estimate for the three missing data points. For ethical reasons the wait list received treatment immediately following the post assessment. Hence, there is no control group data at 6-month follow-up.

Participants who changed their medication or began psychological treatment during the study were considered unchanged in the analysis to control for effects from other treatments (treatment n = 1, control n = 3, NS). Two participants gave inadequate answers on the IPAQ pre-treatment, and were therefore excluded from the analysis on that measure.

Clinically significant change on the main outcome measure, BDI-II, was calculated according to Jacobson & Truax (1991), with test-retest data from Sprinkle et al. (2002), and clinical cut-off scores in accordance with the BDI-II manual (Beck & Steer, 1996). Reliable change (RC) was calculated with 95% confidence intervals.

Results

As evident from Table 1, participants were predominantly female (83%), currently married or cohabiting (46%), had attained college or higher-level education (71%), and had previous experience of psychotherapy (62%).

As evident from Table 1, the treatment- and the control group should be considered equal on all demographic characteristics with all p’s > .07 (for sex χ21 = 0.0, p = 1.0; age t40.66 = 0.25, p = .80; marital status χ23 = 2.92, p = .40; highest educational level χ24 = 4.12, p = .40; medication, χ22 = 5.34, p = .07; and psychotherapy χ22 = 0.0, p = 1.0).

During the treatment period of 9 weeks, 14/24 (58%) of the participants in the treatment group completed all treatment modules (see Fig. 1). The remaining participants did not complete the program within the given time frame, or dropped out. However, in compliance with the ITT principle, no participants were excluded from the statistical analyses due to low adherence or drop-out. Reasons given for not completing the program were that participants believed they did not have sufficient time for physical activity (n = 4), the program was perceived as non-effective (n = 2), or that changes in life events or sickness made it impossible for participants to complete the program (n = 1). Some participants ended the program without giving any reason for termination (n = 3).

Table 3 includes means, effect sizes and tests of significance for group-, time-, and interaction effects on all outcome measures. No significant difference was found at pre-treatment between groups on all measures. Post-testing indicated that the treatment condition was superior to the control on both measures of depression (MADRS-S & BDI-II), with significant interaction effects and medium effect sizes. No significant differences were found between groups on measures of anxiety (BAI), quality of life (QOLI), and physical activity level (IPAQ). Both groups achieved significant decreases in depressive symptoms and anxiety, and increases in quality of life and physical activity level during the study.

Follow-up after 6 months

As seen in both Table 3 and Fig. 2, the treatment effects were maintained at the 6-month follow-up. Actually, on the general measure of anxiety (BAI), there was a continued improvement between post and follow-up (t23 = 2.51, p < .05). The mean post to follow-up within-group effect size was d = 0.17 with a low of 0.04 (IPAQ) and a high of d = 0.35 (BAI).

Figure 2 Mean scores and confidence intervals.

The Beck Depression Inventory II (BDI-II) scores at pre, post and follow-up for the Treatment and the Control group including 95% confidence intervals.

Clinical significance

At post-treatment, 17/24 (70.8%) of the participants in the treatment group were considered reliably improved on the main outcome measure, and 9/24 (37.5%) as both reliably improved and recovered. In the control group, 8/24 (33.3%) were considered improved, and 1/24 (4.2%) reliably improved and recovered (Fig. 3).

Figure 3 Clinically significant change on BDI-II post-treatment.

Discussion

The aim of this study was to develop and evaluate a treatment program for mild to moderate major depressive disorder based on physical activity administered via the Internet. The hypotheses were that participants in the treatment condition would have reduced their depressive and anxiety symptoms and at the same time increased levels of physical activity and quality of life more than participants in the control group.

Results showed a statistically significant interaction effect at post-testing favoring the treatment condition compared to the control condition, with a moderate between-groups effect size of Cohen’s d = 0.67 (95% confidence interval: 0.09–1.25) on the main outcome measure of depression, BDI-II. The within-group effect size was large for the treatment group with Cohen’s d = 0.89 (95% confidence interval: 0.67–1.92) and moderate for the control group with d = 0.62 (95% confidence interval: 0.03–1.19).

The effects found in the current study are in line with efficacy outcomes from other well-established evidence-based psychological treatments. For example, Silveira et al. (2013) reviewed the effects of physical activity for depression and found an effect size of d = 0.61 for depressive symptoms compared to the control in all the studies included in the analysis. In addition to this, Andersson & Cuijpers (2009) found a between-group effect size of d = 0.61 for supported computerized CBT treatments for depression, with a majority being studies on Internet-delivered CBT.

However, results in this study showed no significant difference between the groups on secondary measures of anxiety, physical activity, and quality of life (BAI, IPAQ, and QOLI). A possible explanation for this could be the low statistical power of the study due to a small sample size of N = 48. Calculations made prior to the study indicated that a sample size of N = 80 would be required to find significant interaction effects if they existed in the population. The main reason for the small sample size was in part the limited number of participants registering their interest in the study (N = 159), and in part the large percentage of excluded participants (69.8%). The main reason for this exclusion rate was high self-rated levels of physical activity at baseline for participants registered for the study. Also, these findings are in line with the results in a similar trial where no relation between reduced depressive symptoms and secondary measures such as changes in quality of life was found (Dozois, Dobson & Ahnberg, 1998).

The most surprising findings were the non-significant interaction effect on IPAQ. As noted above, sample size limited power to detect this interaction. A placebo effect in the active treatment group may also explain this result. Various additional explanations can be entertained. Firstly, a large number of the participants in the study showed moderate or high levels of physical activity at pre-treatment (60.5%). This ceiling effect in the sample provided small opportunities for increased levels of physical activity. Ideally, only participants with a low level of physical activity should have been included in the study since those individuals were expected to gain most from the treatment. However, such a rigorous exclusion would have left few participants to the study. Secondly, the IPAQ has several issues in need of consideration. Ekelund et al. (2006) found that people significantly overestimate their physical activity using the IPAQ compared to using an objective measure. This brings some uncertainty to the results found in this study. It is possible that people in the treatment condition overestimated their physical activity less at post-treatment, than people in the control condition, since the treatment included detailed monitoring of physical activity. Thirdly, data cleaning was needed due to unreasonable answers. This was done to answers from eight participants according to principles from the International Physical Activity Questionnaire Group (2005). This process may have led to misinterpretations of the intended answers of the respondents. Lastly, four participants gave incomprehensible answers to questions on the IPAQ, indicating misunderstandings of the instructions.

In summary, the findings in this study indicate that internet-administered therapist-guided physical activity can be an effective treatment for depressive symptoms for people with mild to moderate major depression, but there is no evidence of effectiveness in raising levels of physical activity or quality of life, nor reducing symptoms of anxiety. Since the effects found for depressive symptoms cannot be explained by changes in physical activity, questions are raised concerning the active ingredients in the treatment.

While Internet-delivered therapy has potential benefits there are also issues of potential concerns, e.g., the risk of missing physical signs of depression such as agitation or retardation; difficulty identifying clients adequately for follow-up of increasingly suicidal patients, including involuntary treatment if required.

Some researchers state that there is no dose–response relationship between levels of physical activity and depressive symptoms (Kesaniemi et al., 2001). This implies that other aspects than the frequency, duration, and intensity of physical activity mediates changes in depressive symptoms. Considering this, it seems unlikely that improved fitness and related physiological changes account for reduced depressive symptoms in this study. Rather, it seems that other aspects of the treatment account for the obtained effects.

Firstly, earlier research has shown that self-help programs in which support is provided are more effective than programs without support (Spek et al., 2007; Palmqvist, Carlbring & Andersson, 2007). In this program, feedback based on motivational interviewing principles (cf. Miller & Rollnick, 2002) was given each week. This could possibly explain the positive outcome to some extent.

Secondly, the program included features of behavioral activation strategies such as exercise planning and monitoring. This was introduced early in the program and continued until the end. Research shows that behavioral activation is an effective treatment for depression (Dimidjian et al., 2011) which can be administered over the Internet (Carlbring et al., 2013a). The planning and monitoring of exercise could account for some of the positive outcome effect on depressive symptoms.

Thirdly, a reoccurring feature of the treatment was that participants set their goals independently. Participants were encouraged to set efficient goals every week and to achieve them. Studies have shown increased levels of self-efficacy in people striving for and achieving their goals (Biddle & Fox, 1998). To obtain goals and to enhance self-efficacy for physical activity can be seen as positively reinforced behavior. According to the spiral model, based on the same principles as behavioral activation, this can be a way to break depressive patterns and inactivity (Waller & Gilbody, 2009).

A pedometer was also sent to each participant in the treatment condition. The purpose was to increase adherence to the program and to maximize the likelihood of participants engaging in physical activity. However, it could be argued that the pedometer that was sent free of charge was simply a way of enhancing a possible placebo effect (Sliwinski & Elkins, 2013).

A little more than half (58%) of the participants in the treatment condition completed all nine text modules on time. This low adherence rate is a cause for concern, but is in line with earlier reviews of Internet-based treatment studies which found that just over half of the participants complete all sessions during the treatment period (Eysenbach, 2005).

All participants included in the study completed post-testing, which is uncommon for studies of this kind (Christensen & Mackinnon, 2006). Usually, the last observation carried forward principle is used for missing data. Having a complete dataset for both pre- and post-testing for both conditions increases reliability and validity of the results. Since it is feasible that other active treatments influence outcome measures in this study, a conservative way of handling data was preferred.

There were significantly more females than males in the study. This is common for studies on depression, but should be considered as a limitation for the generalizability of the results. In addition, the mean age in the sample was high (Table 1), with only 10 participants under 40 years of age.

Future research could dismantle the different parts of the treatment, such as physical activity, therapist support, and self-efficacy to estimate to what extent they influence the outcome (cf. Carlbring et al., 2013b). Studies should include objective measures of physical activity as well as measures of mediating factors to distinguish between active ingredients of the treatment.

This study has introduced a new, potentially effective Internet-based treatment for depression based on a physical activity intervention. The treatment program may be a valid alternative to traditional treatments for depression for people unwilling to use antidepressant medication or psychotherapy. Because of the Internet-delivered nature of the program, it may be considered cost-effective and not limited by large geographical distances (Hedman, Ljótsson & Lindefors, 2012).

Thank you to Lina Aittamaa, Linda Ek, Linda Westling, Mikael Granlund, Jessica Henriksson, and Linda Ternedal for serving as therapists, and to Alexander Alasjö for web programming.

Additional Information and Declarations

Competing Interests

Author Contributions

Ethics

Data Deposition

Clinical Trial Registration

Gerhard Andersson is an Academic Editor for PeerJ.

Morgan Ström, Carl-Johan Uckelstam and Per Carlbring conceived and designed the experiments, performed the experiments, analyzed the data, contributed reagents/materials/analysis tools, wrote the paper.

Gerhard Andersson, Peter Hassmén and Göran Umefjord contributed reagents/materials/analysis tools, wrote the paper.

The following information was supplied relating to ethical approvals (i.e., approving body and any reference numbers):

This study was approved by Etikprövningsnämnden i Umeå: 2011-145-31 Ö.

The following information was supplied regarding the deposition of related data:

Dryad Digital Repository: http://doi.org/10.5061/dryad.c6q65.

The following information was supplied regarding Clinical Trial registration:

Registered at ClinicalTrials.gov: NCT01573130.

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
