# Peer review of "Internet-delivered therapist-guided physical activity for mild to moderate depression: a randomized controlled trial"

_PeerJ, doi:10.7717/peerj.178_

## Round 0.1 · original submission · Minor Revisions

· Academic Editor

Minor Revisions

I agree with nearly all the comments of the two reviewers. New data are not required, so the revisions are minor in scope, but revisions to the text are of major importance before the manuscript can be accepted for publication.

It is OK to keep the statement “Physical activity as a treatment for depression can be delivered in the form of Internet-based self-help.” Similarly, you can keep "This study has introduced a new, potentially effective Internet-based treatment for depression based on a physical activity intervention." However, all claims that this study shows that physical activity per se is effective at treating major depression must be eliminated. For instance, the claim "physical activity is effective for depressive symptoms for people with major depression" is not supported by the data.

Add a short statement to the Discussion about potential concerns with Internet-delivered therapy (e.g. missing physical signs of depression such as agitation or retardation; difficulty identifying clients adequately for follow-up of increasingly suicidal patients, including involuntary treatment if required).

Please add the rationale for including anxiety and quality of life as outcomes (don't eliminate them from the report).

Please check, but the between-group effect size of 0.61 in the abstract appears to be an error. Also, the last two confidence intervals in the 2nd paragraph of Discussion are identical, also probably an error.

Please change "amount of participants" to "number of participants". Also, "free of change" should read "free of charge."

Reviewer 1 ·

Basic reporting

The study addresses and interesting topic which is of relevance. The manuscript seems to adhere to the all PeerJ policies, is written well, conforms to the PeerJ templates, includes appropriate tables and figures, and is self-contained. I would suggest, however, that the authors could improve their introduction to better demonstrate how their work fits into the broader field of knowledge. Specifically, the introduction could be improved by including a better description of what mechanics or theories of behaviour change they utilized in an effort to change physical activity in their intervention (and why they chose those ones), and how/why it was anticipated that physical activity would change depression.

Experimental design

The manuscript presents an original primary research study within the Journal scope. The research question was clearly defined and the research was conducted in conformity with prevailing ethical standards of the field. Suggestions to improve the research question and the clarity of the methods section follows: (1) The objective listed in the abstract needs to state the objective of the study, rather than summarize the background literature. (2) It was unclear why anxiety and quality of life were included as outcomes. Please either add rationale for these outcomes in the introduction or exclude them for analyses. (3) Please include the information about the study drop-out during treatment (described on page 6) in the Figure 1. (4) It was unclear from this description of the study drop-out how many of the study participants completed some, but not all, of the modules and why they were excluded from the analyses. (5) More information is needed about the procedures applied in delivering the intervention (how did it all work), and more info is needed about intervention content and modules as well. Perhaps the authors could provide examples of some excerpts of the text? At present if would be very hard to replicate this study on the basis of the information provided.

Validity of the findings

The data and analyses seemed reasonably sound and controlled, but suggestions for bettering the conclusions follow: (1) The authors should give a rationale for not included this group in the 6 mo. follow-up analyses. (2) A few times, the authors make conclusions beyond the scope of the results of their data. For example, in the abstract they state, “Physical activity as a treatment for depression can be delivered in the form of Internet-based self-help.” Be it that physical activity was not found to differ between the control and treatment group, it seems unjust to make such claims. The authors should take consideration to adjust conclusions throughout the manuscript. (3) In the conclusion section, they refer to physical activity as a secondary outcome measure. This seems odd based on the study rationale that physical activity is the mechanism of depression change. (4) More should be included in the conclusions about what specifically about their study may have lead to a lack of sig. Between-group effect of physical activity behaviour change. As is, they list general rationales that may apply to most interventions. (5) The authors suggest that a physical activity ceiling effect may explain the lack of sig. between-group change, but physically active people were screened out prior to data collection. This rationale and a more specific description of the screening process is needed.

Reviewer 2 ·

Basic reporting

Please include the basic inclusion criteria for the study in addition to the exclusion criteria.

Experimental design

As written, the intervention is not described in enough detail to be reproduced. An overview of the specific content covered in each of the modules is warranted as is a description of what the participants were encouraged/expected to do each week. Concrete examples of the “weekly planning of exercise” and “home assignments” and therapist-participant interactions would be helpful.

The first paragraph of the Discussion states that a correlation was expected between changes in depressive symptoms and changes in physical activity. This hypothesis was not stated earlier in the manuscript, nor was it tested statistically. This statement should either be removed from the Discussion or, preferably, addressed using available data.

Related to the above point, the provision of a pedometer is mentioned in the Intervention section, but there is no information regarding its purpose or use. If the pedometer isn’t relevant to the study, it should be removed from the report. If it is an important component of the intervention, its purpose should be described in the Methods section. Moreover, if pedometer data are available, it seems like they would be useful in addressing a number of issues (particularly in light of the problems with the IPAQ), such as whether the intervention affected physical activity levels or whether changes in physical activity were associated with changes in depressive symptoms.

Please report the results of the sample size calculations for the primary outcomes in the Methods section. From the Discussion, it appears that these analyses were conducted for the secondary outcome measures, so I assume they were also done for the primary measures. This information should be reported.

Validity of the findings

This statement in the discussion is untrue and contradictory: “In summary, the findings in this study indicate that physical activity is effective for depressive symptoms for people with major depression, but there is no evidence of effectiveness in raising levels of physical activity.” The study did NOT indicate anything about the effects of physical activity on depressive symptoms because there were no changes in physical activity and because there was no demonstration of a relationship between physical activity and depressive symptoms. Instead, the study indicated that participating in the internet-based self-help intervention improved depressive symptoms; however, as the authors note, the mechanism for this effect is unknown and appears NOT to be related to physical activity (at least as measured by IPAQ).

---

## Round 0.2 · accepted · Accept

· Academic Editor

Accept

Thank you for addressing the concerns raised by the reviewers. I agree that your changes have improved the quality of the manuscript.

In the revision you perhaps inadvertently removed from the discussion your previous mention of placebo effects. I believe it would further improve the manuscript to reinstate it, especially since the control condition is a wait list rather than a placebo treatment. I suggest one possible solution here:

"The most surprising findings were the non-significant interaction effects on IPAQ. [INSERT] As noted above, sample size limited power to detect this interaction. A placebo effect in the active treatment group may also explain this result. Various additional explanations can be entertained. [END] Firstly, a large number ..."

Please notify the production staff of your decision on this suggested change.

Nevertheless, since this is a small change I chose "accept" rather than "minor revisions".